# Co-Extraction of Flaxseed Protein and Polysaccharide with a High Emulsifying and Foaming Property: Enrichment through the Sequence Extraction Approach

**DOI:** 10.3390/foods12061256

**Published:** 2023-03-16

**Authors:** Kang-Yu Li, Jie-Ting Ye, Jing Yang, Jia-Qi Shao, Wei-Ping Jin, Chang Zheng, Chu-Yun Wan, Deng-Feng Peng, Qian-Chun Deng

**Affiliations:** 1Key Laboratory of Oilseeds Processing, Ministry of Agriculture, Oil Crops Research Institute, Chinese Academy of Agricultural Sciences, Wuhan 430062, China; lkyfoods0421@163.com (K.-Y.L.); jietingy@163.com (J.-T.Y.);; 2College of Food Science and Engineering, Wuhan Polytechnic University, Wuhan 430023, China; 3Hubei Research Center of Oil and Plant Protein Engineering Technology, Oil Crops and Lipids Process Technology National & Local Joint Engineering Laboratory, Wuhan 430062, China

**Keywords:** flaxseed, plant protein, co-extraction, polysaccharide, composition, structure, functionality

## Abstract

A new focus with respect to the extraction of plant protein is that ingredient enrichment should target functionality instead of pursuing purity. Herein, the sequence aqueous extraction method was used to co-enrich five protein-polysaccharide natural fractions from flaxseed meal, and their composition, structure, and functional properties were investigated. The total recovery rate of flaxseed protein obtained by the sequence extraction approach was more than 80%, which was far higher than the existing reports. The defatted flaxseed meal was soaked by deionized water to obtain fraction 1 (supernatant), and the residue was further treated to get fraction 2 (supernatant) and 3 (precipitate) through weak alkali solubilization. Part of the fraction 2 was taken out, followed by adjusting its pH to 4.2. After centrifuging, the albumin-rich supernatant and precipitate with protein content of 73.05% were gained and labeled as fraction 4 and fraction 5. The solubility of fraction 2 and 4 exceeded 90%, and the foaming ability and stability of fraction 5 were 12.76 times and 9.89 times higher than commercial flaxseed protein, respectively. The emulsifying properties of fractions 1, 2, and 5 were all greater than that of commercial sodium caseinate, implying that these fractions could be utilized as high-efficiency emulsifiers. Cryo-SEM results showed that polysaccharides in fractions were beneficial to the formation of network structure and induced the formation of tighter and smoother interfacial layers, which could prevent emulsion flocculation, disproportionation, and coalescence. This study provides a reference to promote the high-value utilization of flaxseed meals.

## 1. Introduction

Meeting the increasing global demand for protein while keeping sustained development and protein supply is a challenge for the food industry. The consumption of large amounts of animal protein will bring environmental pressure and have potential impact on biodiversity as well as the carbon and nitrogen cycle. Hence, excavating alternative proteins, including those derived from plants, microorganisms, marine organisms, and insects, will meet the needs of global population growth [1]. Obviously, plant protein is the optimal solution because of its mature supply chain, established consumer acceptability, and high nutritional value [2]. However, compared with animal protein, plant protein with high functional properties is difficult to enrich and shows limited application in food ingredients.

In recent years, a new trend in plant protein research is that the enrichment of plant based-ingredients should target functional properties but not purity. In the traditional plant protein enrichment paradigm, harsh extraction conditions are adopted to achieve higher purity, including the extensive use of water, organic reagent, strong acid/alkali, physical stresses, etc. [3]. Furthermore, increased purity of plant protein is at the expense of yield for most extraction methods [4]. On the contrary, some studies have shown the functionalities of some plant protein products, containing non-protein parts, are equal to or even better than a highly purified protein isolate [5]. Geerts et al. [6] found that the residual starch particles in pea protein extracted by aqueous fractionation method would swell in the emulsification process to enhance the stability of the protein-covered emulsion. Another study indicated that the foamability and stability, respectively, of mildly prepared Bambara groundnut protein containing more lipids, phenols, and starch were 9 times and 8 times higher than those of extensively extracted proteins (including dehulling, defatting, and milling-based) [7]. A new notion, targeted co-extraction of ingredients with high functionality from natural biomass through a mild, green, simple, and efficient method, should be increasingly concerned. Proteins and polysaccharides, as the two basic nutrients in organisms, exist widely and stably in most biomass. Many papers have reported that the combination of protein and polysaccharide could benefit their functional properties (such as solubility, emulsifying, as well as foaming properties) and expand their application scope [8,9,10,11]. Based on the above two points, it is of great significance to focus on co-extracting natural fractions rich in protein and polysaccharide from biomass and to study their possible synergistic mechanisms for functional properties.

Flaxseed is an important oilseed crop, which can be used as a sustainable source of protein in addition to oil. The whole flaxseed contains 18~30% protein, 20~35% polysaccharide, 30~41% fat, and 3~4% ash, and its proximate composition is quite similar to soybean [12], in which its defatted meal has been developed as value-added protein. However, the defatted flaxseed meal has long been utilized as a low-value protein fortifier in animal feed or fertilizers, although its protein content exceeds 35% [13]. Studies demonstrated that polysaccharides in the seed coat of flaxseed blocked the separation of proteins, because of their swelling in aqueous media [14]. This further caused the consideration of cost in the industrial production of high-purity flaxseed protein. Besides, compared with the commonly utilized polysaccharide in food ingredients, such as carob gum, guar bean gum, and xanthan gum, flaxseed polysaccharide is considered to be healthier [15]. Nikbakht Nasrabadi et al. [16] found that the artificial use of flaxseed protein and flaxseed gum could improve their emulsifying and foaming properties. Inspired by this, it is possible to enrich flaxseed protein without separation from flaxseed polysaccharide. Thus, flaxseed may be an ideal model to realize the co-extraction of protein-polysaccharide natural mixtures with high functionality for food industry application.

The purpose of the current work was to co-extract protein-polysaccharide natural fractions with high functional properties, such as solubility, as well as emulsifying and foaming properties, from flaxseed meal. Here, we built a sequence aqueous extraction process, which is a green and mild extraction method. Five fractions rich in protein and polysaccharides with different proportions were co-extracted from the defatted flaxseed meal through water fractionation, which covered the whole flaxseed meal. In order to guide the targeted application of these fractions, their composition (including protein and polysaccharide), structure, and functional properties were studied and compared with the most commonly used commercial proteins in the food industry.

## 2. Materials and Methods

### 2.1. Materials

Flaxseed (variety: zhangya 2#) was provided by the Gansu Academy of Agricultural Sciences (Lanzhou, China). Flaxseed oil was bought from the Beijing Red Well source Trading Company (Beijing, China). Commercial sodium caseinate (C-SC, protein content ~90.56%) was obtained from Sigma Aldrich (St. Louis, MO, USA). Commercial soy protein concentrate (C-SPC, protein content ~83.23%) was obtained from Linyi Shansong Biological Products Co., Ltd. (Linyi, China). Commercial pea protein concentrate (C-PPC, protein content ~80.47%) was purchased from Hengyuan Biological Products Co., Ltd. (Zaozhuang, China). Commercial flaxseed protein concentrate (C-FPC, protein content ~83.39%) was obtained from Xi’an Guohao Biological Products Co., Ltd. (Xi’an, China). Other chemicals and reagents were of analytical grade.

### 2.2. Co-Extraction of Protein-Polysaccharide Natural Fractions from Flaxseed

Five flaxseed protein-polysaccharide natural fractions varying in protein and polysaccharide content were obtained by sequence aqueous extraction processes. Figure 1 showed a schematic overview of the co-extraction processes.

The cold-pressed flaxseed meal was gained by cold pressing (CA59G, Komet, Detmold, Germany), followed by crushing using a grinder (800A, REDSUN, Jinhua, China) and refining by a 60-mesh sieve. After defatting by Soxhlet extraction using petroleum ether, the defatted meal was dissolved in warm deionized water at 40 °C in a ratio of 1:15 and stirred for 3 h without adjustment of pH. The dispersion was centrifuged at 10,000× *g* for 30 min at 20 °C, and then, the supernatant was freeze dried to gain fraction 1. The precipitate was re-dispersed in water to the same volume as the previous step and stirred for 2 h at pH 9.0. After centrifugation, the freeze-dried supernatant and precipitate were labeled as fraction 2 and fraction 3, respectively. Part of the supernatant (fraction 2) was further separated by isoelectric precipitation at pH 4.2. The acid dispersion was then centrifuged and the freeze-dried supernatant was marked as fraction 4. Finally, the precipitate was re-dispersed in water and then neutralized to pH 7.0 using 1.0 M NaOH. The freeze-dried precipitate was named fraction 5.

### 2.3. Component Determination of the Fractions

The fraction was continuously heated in a muffle furnace at 550 °C; the ratio of residual and fraction mass represented the ash content. The protein content was determined by the Kjeldahl method using DigiPREP TKN Systems (Hanon K9860, Jinan, China). The nitrogen conversion factor was 6.25. Moisture content was the ratio of the moisture volatilized at 105 °C to the sample weight. Total carbohydrate was calculated as the total mass of the sample minus that of other components.

### 2.4. Extraction Rate and Protein Recovery

The extraction rate of fractions was calculated by Formula (1):(1)$$\mathrm{Extraction}\;\mathrm{rate}\;(\%)=\frac{m_{f}}{m_{d}}\times 100$$

The protein recovery of fractions was calculated by the following Formula (2):(2)$$\mathrm{Protein}\;\mathrm{recovery}\;(\%)=\frac{c_{f}\times m_{f}}{c_{d}\times m_{d}}$$
where *m_f_*, *m_d_*, *c_f_*, and *c_d_* represent the mass of fraction, mass of defatted flaxseed meal, protein content of fraction, and protein content of defatted flaxseed meal, respectively.

### 2.5. Amino Acid Composition Profile

A total of 80 mg of freeze-dried fraction was put into 10 mL of 6 mol/L HCl; then, it was sealed after nitrogen blowing for 30 s and hydrolyzed at 110 °C for 48 h, and the hydrolyzed solution was filtered with a 0.45 μm filter membrane and diluted to 50 mL. Then, 2 mL diluted solution was taken out after constant volume, which was put on a rotary evaporator for deacidification at 45 °C until a little solid or stain was left at the bottom of the bottle. Finally, 2 mL of sample buffer solution was added to fully dissolve it, which was filtered through a 0.45 um filter. The amino acid composition was determined by an amino acid analyzer (Biochrom30+, Cambridge, UK).

### 2.6. Sodium Dodecyl Sulfate–Polyacrylamide Gel Electrophoresis (SDS-PAGE)

SDS–PAGE was carried out based on the method reported by Yang et al. [17] with minor modifications. The solution of freeze-dried fraction (2 mg/mL) was added to a 10 μL sample dissolve buffer (3% sodium dodecyl sulfate, 10% glycerin, 0.05% bromophenol blue). Then, the mixture was bathed in boiling water for 5 min. Next, 10 μL of mixture was added to the gel pore, including 12% separating gel and 5% stacking gel. Gels were run at 70 V for 30 min and 110 V for 90 min before staining with Coomassie brilliant blue R-250.

### 2.7. Carbohydrate Composition Profile

In total, 5.0 mg of sample was hydrolyzed using 1 mL 2 M trifluoroacetic acid at 121 °C for 2 h. Using nitrogen gas and methanol to remove the excess trifluoroacetic acid, the fraction solution was injected into the high-performance anion-exchange chromatography with pulsed amperometric detection (HPAEC-PAD), which consisted of an ICS-5000 system (Thermo Fisher Scientific, Waltham, MA USA) equipped with Dionex™ CarboPac™ PA20 (150 × 3.0 mm, 10 μm, Thermo Fisher Scientific, Waltham, MA, USA).

### 2.8. ζ-Potential and Particle Size Distribution

The 1 mg/mL fraction solution was poured into special measuring containers, and ζ-potential and volume particle size distribution of the fraction were analyzed by Zetasizer Pro (Malvern, UK).

### 2.9. Differential Scanning Calorimeter (DSC)

Around 5 mg dried fraction was packed in an aluminum crucible, and heated from 25 °C to 160 °C with a speed of 10 °C/min to determine the denaturation temperature (*T_d_*) using a differential scanning calorimeter (Mettler, Greifensee, Switzerland). An empty aluminum crucible was used as a control.

### 2.10. Solubility

First, 1% (*w*/*v*) flaxseed fraction solution was dissolved in deionized water for 4 h under magnetic stirring. Next, the solution was centrifuged at 7200× *g* for 10 min and precipitate was removed. The freeze-dried supernatants were weighed. The soluble component was divided by the initial mass to express solubility. The measurement result of C-FPC solution under the same conditions was used as a control.

### 2.11. Emulsifying Properties

The 1% fraction solution was adjusted to pH 7.0, and 18 mL of this solution was mixed with 2 mL flaxseed oil using a high-speed disperser (IKA, T25, Wilmington, NC, USA) at 13,000 rpm for 2 min. The resultant emulsion (50 μL) was added to 5 mL of 0.1% SDS solution. Next, the absorbance of the mixture was measured at 500 nm (*A*_0_). The emulsion was set at 25 °C for 10 min and the absorption was measured by previous steps (*A*_10_). The emulsifying activity index (EAI) and emulsifying stability index (ESI) were calculated by Formulas (3) and (4): (3)$${\mathrm{EAI}\;(m}^{2}/g)=\frac{2\times {2.303\times A}_{0}\times \mathrm{DR}}{C\times \theta \times 10,000}$$
(4)$$\mathrm{ESI}\;(\min)=\frac{A_{0}\times 10}{A_{10}{-A}_{0}}$$
where *DR*, *C*, and *θ* represent the dilution rate, protein concentration, and oil volume fraction, respectively. The measurement results of C-SC, C-SPC, C-PPC, and C-FPC solutions under the same condition were used as control.

### 2.12. Interfacial Microstructure of Emulsion

The microstructure of the resultant emulsions stabilized by fractions was observed using cryo-scanning electron microscopy (cryo-SEM) (SU8000, Hitachi, Tokyo, Japan). A total of 5 μL of emulsion was placed in liquid nitrogen and frozen, and then cut in the cryo-preparation chamber. Next, its cross-section was sublimated at −70 °C for 12 min. After the sputter-coating for 12 s, the interface microstructure and continuous phase of emulsion were analyzed by cryo-SEM.

### 2.13. Foaming Properties

Here, 15 mL of 1% fraction solution was added into a glass bottle and a high-speed disperser was utilized to disperse the solution at 13,000 rpm for 2 min. Then, the resultant foam was immediately transferred to a 50 mL measuring cylinder. The volume of foam stored at room temperature for 2 min (*V*_2_) and 30 min (*V*_30_) was measured. Foaming ability (FA) and foaming stability (FS) were calculated by Formulas (5) and (6):(5)$$\mathrm{FA}\;\left(\%\right)=\frac{V_{2}}{15}\times 100$$
(6)$$\mathrm{FS}\;\left(\%\right)=\frac{V_{30}}{V_{2}}\times 100$$

The measurement results of C-SC, C-SPC, C-PPC, and C-FPC solutions under the same condition were used as control.

### 2.14. Statistical Analysis

The standard deviation of the mean value was used as a measure of the error. IBM SPSS Statistics 25 (IBM, Chicago, IL, USA) was used to apply one-way ANOVA using the post-hoc Duncan method. Significance was defined as *p* < 0.05.

## 3. Results and Discussion

### 3.1. Co-Extraction of Five Fractions and Their Composition

The effect of each extraction step on the composition of flaxseed protein-polysaccharide natural fractions was investigated (see in Table 1). The first step was to extract fraction 1 by simple water soaking, and the protein and polysaccharide content was 46.81% and 45.60%, respectively. Interestingly, the ratio of protein and polysaccharide was similar to that of defatted flaxseed meal. It is reported that the soluble carbohydrate in flaxseed accounts for ~31% of its total carbohydrate content [18]. Based on this ratio, more than 90% of the soluble carbohydrate was extracted through the water washing process. Fraction 2 was obtained by a simple solubilization step at pH 9.0, and it contained 59.84% protein and 28.63% carbohydrate. The protein content of the separated precipitate (fraction 3) was 37.45%, which contained most of the insoluble protein and polysaccharide in defatted flaxseed meal. To obtain commercial flaxseed protein concentrate with higher purity, the solubilized extract at pH 9.0 was treated by acid precipitation. The freeze-dried supernatant and the re-dissolved precipitate were labeled as fraction 4 and fraction 5, respectively. The protein content of fraction 5 was as high as 73.05%, which was far higher than the standard value of flaxseed protein concentrate (>60%). It was also higher than the flaxseed protein concentrate extracted from the whole flaxseed meal by Lan et al. [12]. It is worth mentioning that in the optimization of acid precipitation pH, our team found that the isoelectric point of flaxseed protein shifted, mainly because of the interaction between flaxseed protein and polysaccharide, which led to charge neutralization [19]. 

These two processes (fraction 1 + fraction 2 + fraction 3; fraction 1 + fraction 3 + fraction 4 + fraction 5) caused the total extraction rate and total protein recovery rate of the defatted flaxseed meal to exceed 90% and 80% respectively. Fraction 1, fraction 2, and fraction 5 had better emulsifying or foaming properties than commodity protein, which could be verified in the following results. The total extraction rates of the two processes of the three fractions were 48.55% and 38.72% respectively, and the total protein recovery rates were 57.93% and 46.50% respectively. The extraction rate and protein recovery rate of flaxseed protein concentrate extracted from flaxseed meal by Lan et al. [12] were only 11.5% and 24.7%, respectively. The protein recovery rates of flaxseed protein isolate extracted by Kaushik et al. [19], after flaxseed was degummed at different temperatures, were 12.1~20.29%. Therefore, compared with the traditional protein extraction method, the milder co-extraction method could effectively enhance the utilization rate of flaxseed meal to increase its added value.

### 3.2. Protein Characterization

#### 3.2.1. Amino Acid Composition Analysis

The amino acid composition of protein affects its nutritional value, structure, and functional properties. As can be seen in Table 2, 17 basic amino acids are measured. It was reported that flaxseed protein isolate exhibited a preferable amino acid composition and contained proper essential amino acid content, comparable with soy protein isolate and pea protein isolate [19]. Table 2 also showed that glutamic acid was the major amino acid in the five fractions. Fraction 4 exhibited only one band in SDS-PAGE image (Figure 2), which represented the albumin subunit. This phenomenon indicated that flaxseed albumin might contribute more glutamic acid than other types of proteins. In addition, the amino acid profile of the five fractions were quite different, which was related to protein type, solvent polarity, the destruction degree of subcellular structure by treating conditions, and the effect of non-protein parts on the dissolution of protein. Further research could be carried out by more advanced means, such as mass spectrometry, size exclusion chromatography, etc. Nwachukwu and Aluko [20] found that flaxseed albumin possessed higher negative charge amino acids, while globulin contained more sulfur-containing amino acids and hydrophobic amino acids. Fraction 1 and 4 contained higher negatively charged amino acids, indicating that more albumin was dissolved in the steps of water soaking and acid extraction, which was consistent with the results in Figure 3. The content of hydrophobic amino acids in fraction 2 was the highest compared with other fractions, as high as 455.76 mg/g protein, which might be beneficial for their adsorption and rearrangement process at the oil–/air–water interfaces [21]. After acid precipitation, fraction 5 had the reduced content of sulfur-containing amino acids and hydrophobic amino acids, which was because a small amount of these two amino acids exists in acid-soluble fraction 4 (mainly albumin). It is worth mentioning that aromatic amino acids mainly existed in fraction 3 and 5, and had the lowest content in fraction 4. This indicated that they could be enriched in flaxseed alkali insoluble protein and globulin. 

#### 3.2.2. SDS–PAGE

As can be seen in the SDS–PAGE profiles (Figure 2), defatted flaxseed meal showed four predominant bands at 10 kDa, 18~20 kDa, 21~23 kDa, and 30~34 kDa, respectively, and a minor band at 55 kDa. It was reported that the two predominant bands at 10 kDa and 46~55 kDa could be assigned to albumin (1.6–2 S) and main band of globulins (11–12 S), respectively, which was consistent with the result of Figure 2 [14,20]. In addition, Marcone et al. reported that [22] the 30.0 and 35.2 kDa bands corresponded to acidic subunits, while the 24.6 kDa band represented the basic subunit. In another paper, the acidic subunits (30~39 kDa) were distinguished from the basic bands with molecular weight between 20 and 27 kDa [23]. The conclusions of the above reports were similar to the band distribution between 17~43 kDa in Figure 2.

The band intensities differed because the fractions varied in protein content. However, the distribution of bands of fractions 1, 2, and 5 was similar. Compared with the band of fraction 1, the bands of fractions 2 and 5 were deeper at 30–34 kDa and closer to the intensity of defatted flaxseed meal, indicating that the alkali dissolution process had a more profound impact on the dissolution of acidic protein subunits with larger molecular weight. The band of fraction 3 at 55 kDa was the most obvious, followed by fraction 5. The band composition of SDS–PAGE profiles of fraction 5 was similar to that of flaxseed protein concentrate extracted by Lan et al. [12], except that their samples displayed a very dark color at 55 kDa, which might be due to the differences in flaxseed varieties and planting regions. Interestingly, fraction 4 lacked most of the globulin subunits and only exhibited one band at 10 kDa, which represented the low molecular weight albumins. This is the same as the single band phenomenon shown in the supernatant of yellow pea protein extracted by Kornet et al. [5], using the isoelectric point separation method. Besides, the flaxseed albumin isolated by Nwachukwu and Aluko [20] also displayed a single band at 10 kDa. Kajla et al. [24] found that flaxseed albumin had higher solubility under acidic conditions, while globulin was more soluble under alkaline conditions. This characteristic led to the enrichment of high-purity albumin in fraction 4. This method was expected to become a valid mean to extensively obtain high-purity flaxseed albumin. However, due to the high content of carbohydrate and ash in fraction 4, further purification might be required in the later processing, through membrane separation, ion chromatography and other means. It was also noticed that the minor differences between the fractions and the absence of a band indicate that the co-extraction process did not have a significant impact on flaxseed protein compositions. To sum up, flaxseed proteins with large molecular weight were enriched during alkali dissolution, and fraction 5 had a similar protein composition to the reported flaxseed protein concentrate.

### 3.3. Carbohydrate Analyze

Flaxseed contains ~30% carbohydrate, mainly in the form of polysaccharide, which is divided into soluble carbohydrate and insoluble carbohydrate [25]. The presence of polysaccharides can influence protein functionality, dependent on their content, composition, polymerization, and degree of complexation. Here, the monosaccharide composition of fractions is analyzed and the results are displayed in Table 3.

Glucose was the most major monosaccharide form for fractions 1, 2, 4, and 5, which was in accord with the results reported by Adorian et al. [26], in which the content of glucose in flaxseed meal was the highest. This monosaccharide composition was different from some reports, mainly because the composition of flaxseed polysaccharide from different regions and gene sources showed a difference [27]. In fact, flaxseed soluble polysaccharides are a hetero-biopolymer dwelling of acidic and neutral monomers. Galactose, xylose, glucose, and arabinose were the neutral monomers of flaxseed polysaccharides, whereas fucose, rhamnose, and galacturonic acid represented the acidic monomers [28]. Since neutral polysaccharides contributed more viscosity, it could be speculated that fraction 1 with higher neutral monomer content might have a higher viscosity. Moreover, 1.34% mannose and 0.28% glucuronic acid were detected in fraction 3, while no rhamnose was detected, which proved that there was a significant difference between components of soluble and insoluble flaxseed polysaccharides.

### 3.4. ζ-Potential and Particle Size Distribution

The ζ-potential can reflect the stability of the system. It is generally believed that the higher absolute value of the ζ-potential, the higher the stability of the system [10]. As shown in Figure 3A, fraction 5 had the highest absolute value, followed by fraction 2, fraction 1, and fraction 4. Fraction 3 had the lowest absolute value. The low ζ-potential of fraction 3 might be ascribed to its high degree of aggregation, which results in its charged group being wrapped inside the aggregate.

The particle size distribution of the co-extracted fractions could reflect the degree of aggregation and affect their functional properties. For fraction 3, we did not capture its data, because its large size and serious aggregation were beyond the test range of the instrument. Fraction 1 showed two peak distributions, and the average volume mean was about 742.99 nm (Figure 3B). The particle size of fraction 2 further extracted by alkali decreased significantly to 307.91 nm, and it displayed single peak distribution. This might be due to the depolymerization of aggregates during alkaline treatment. After further acid treatment, the protein re-aggregated near the isoelectric point, and this aggregation process was more profound in fraction 5, which is enriched with more globulin. In this study, a small peak of fraction 1 and fraction 5 exceeding 4000 nm proved the existence of large aggregates. The formation of aggregates might be a result of globulin–globulin hydrophobic interactions, which were improved due to the partial unfolding of the protein after the alkali dissolution [5]. Notably, protein aggregation caused by acid precipitation was difficult to eliminate, even if it was re-dispersed at neutral conditions. It is to be expected that this phenomenon will cause the protein extracted by alkali dissolution and acid precipitation to show poorer functional properties. 

### 3.5. Thermal Stability

As shown in Appendix A, the denaturation temperature (*T_d_*) of the five fractions ranged from 98.81 to 118.69 °C, which were much higher than the reported *T_d_* of flaxseed protein isolate (about 82.52 °C) [29]. This reflected that this co-extraction method could effectively improve the thermal stability of protein from flaxseed, when compared with the traditional alkali solution and acid precipitation method.

### 3.6. Functional Property

#### 3.6.1. Solubility

Solubility is an important parameter related to the functional properties of protein. For example, high solubility can promote surface adsorption of proteins to occur in emulsification or foaming processes [30]. As can be seen in Figure 4, the solubility of C-SC was 100%, and its aqueous solution was clear and transparent, which is also an important reason why it is utilized widely in the food industry. However, the solubility of the three plant-based commercial proteins was generally poor, and the best value was only 35.85% of C-SPC. Except for fraction 3, the solubility of other fractions was much higher than that of C-SPC, CPC, and C-FPC. For example, the solubility of fraction 4 was up to 95.52%, partly due to the contribution of soluble polysaccharides in the fraction. In addition, the traditional alkali dissolution and acid precipitation process would lead to irreversible changes in the advanced conformation of proteins [31,32]. Such changes induced a hydrophobic interaction and the formation of large aggregates, thus reducing solubility. This led to the solubility of fraction 5 being lower than that of fractions 1 and 2. Fraction 3, as the residue of alkali dissolution, was mainly composed of insoluble flaxseed protein aggregates and flaxseed fiber, and its solubility in a neutral environment was measured to be as low as 10.55%. Besides, the reasons for the difference in solubility of different fractions also included the ratio of protein to soluble polysaccharides in the system, the ratio of albumin to globulin in the fractions, as well as non-covalent interactions between substances in the system, etc. [33]. Based on the results displayed in Figure 4, co-extraction of protein and polysaccharide from plants might be a potential strategy to produce high-quality food ingredients.

#### 3.6.2. Emulsifying Properties

Figure 5 shows the EAI and ESI of fractions and control. The EAI of the five fractions was higher than that of C-FPC, C-SC, and C-PPC. The EAI of fraction 1 with the medium protein content was the highest, 1.10 times that of C-SPC and 2.38 times that of C-FPC. In addition, the EAI of fraction 2 was second only to that of fraction 1, and there was no significant difference between this value and the C-SPC, which might be due to its higher hydrophobic amino acid content, high solubility (~91.00%), and smaller particle size.

A stable emulsion system is an important factor to ensure commodity sales. As shown in Figure 6B, except C-SC, other commercial proteins showed a lower ESI, which was why it was difficult to use a single protein as an emulsifier to stabilize emulsion in industrial production. However, the ESI of fraction 1, fraction 2, and fraction 5 was higher than that of C-SC, which was an excellent result. Nwachukwu and Aluko [20] reported that the ESI of flaxseed globulin was higher than flaxseed albumin at pH 7.0 with a concentration of 1%. However, this result was not consistent with that in Figure 6B, where the ESI of fraction 5 enriched with more globulins was lower than that of fraction 1 and 2, illustrating flaxseed polysaccharide played an important role in maintaining the stability of the emulsion.

#### 3.6.3. Interfacial Microstructure of Emulsions Covered by Fractions

We further explored the synergistic adsorption behavior of the co-extracted protein-polysaccharide natural fractions at the oil–water interface and its mechanism of maintaining the stability of the emulsion. The microstructure of the emulsions stabilized by three fractions is observed by cryo-SEM and the results are shown in Figure 6. Filamentous structure was observed in the continuous phase of emulsion systems, which was attributed to the free flaxseed soluble polysaccharide. A similar structure also existed in the continuous phase of emulsion stabilized by whey protein added with pectin [34]. The content of polysaccharide in the fractions seemed to have a significant influence on the structure in the continuous phase. A dense network structure was formed in the continuous phase of the emulsion stabilized by fraction 1 with higher polysaccharide content, bridging between the droplets. The network structure in the continuous phase of the emulsion stabilized by fraction 2 with medium polysaccharide content obviously weakened, and some filaments were connected to the droplets only at one end. When the polysaccharide content in the fraction was lower, the bridge formed by this filamentous structure disintegrated, and more of it was freed in the continuous phase. The influence of polysaccharide on the continuous phase of emulsion is shown in Figure 7. Many studies demonstrated that the network structure formed by polysaccharides in emulsion could support the system and restrict the Brownian motion of emulsion droplets [35].

Furthermore, the impact of flaxseed polysaccharides at the oil–water interface should not be ignored. As shown in Figure 6, the emulsions stabilized by fraction 1 and 2 exhibited a smooth and compact interfacial layer. On the contrary, a large number of folds and depressions were observed on the interface with respect to the emulsion stabilized by fraction 5, which was a typical feature of the protein-stabilized emulsion, due to the weaker elasticity of the protein-forming interface [17,34,36,37,38]. The above results indicated that flaxseed polysaccharides could be continuously adsorbed and filled at the interface of emulsion, forming a stronger elastic interfacial film. This elastic interface can not only inhibit the coalescence and coarsening of droplets, but also expand the application fields of emulsion, such as microcapsules [39].

#### 3.6.4. Foaming Properties

FA and FS are crucial indexes to characterize the foaming properties of materials [40]. As shown in Figure 8A,B, compared with other commercial proteins, the FA and FS of C-FPC are very low, only 18.0% and 6.84%. Although the foaming properties of flaxseed protein extracted by alkali solution and acid precipitation from different varieties may be quite different. After comparing three studies on the foaming properties of flaxseed protein, an obvious conclusion is that the foaming properties of flaxseed protein alone under neutral conditions are poor [12,41,42]. Furthermore, FA and FS of the five fractions were far higher than that of C-FPC. Among them, the foaming property of fraction 5 was the best, and its FA and FS were up to 229.68% and 67.65%, respectively, which were about 12.76 times and 9.89 times higher, respectively, than that of C-FPC. Compared with other commercial proteins, the foaming property of fraction 5 was slightly lower than C-SC, similar to C-SPC, and its FA was 2.34 times higher than C-PPC.

Generally, proteins possess better foaming properties than polysaccharides. Nikbakht Nasrabadi et al. [16] found that although the unique protein structure in flaxseed gum gave it a relatively limited FA, it was only one-seventh of flaxseed protein under the same conditions. In our study, the fractions with higher protein content displayed higher FA, but that of fractions 1 and 3 was contrary to this trend. In fact, the polysaccharide profile in the fraction had an important influence on its foaming properties. Compared with other fractions, fraction 1 contained more neutral polysaccharides (Table 2), which might endow it with higher viscosity, leading to the slow adsorption at the air–water interface. This characteristic might promote the formation of a biopolymer network set by flaxseed gum, which could inhibit the coalescence of air bubbles, and hence, improve the foam stability. Fraction 3, with low solubility and high degree of aggregation, exhibited a surprising foaming property. Its FS reached 49.6%, second only to fraction 1 and 5, and close to C-SPC. We analyzed this phenomenon of fraction 3 and obtained the following speculation. After the alkaline insoluble precipitation was re-dispersed at pH 7.0, its protein and polysaccharide with high hydrophobicity formed hydrated particles, which were irreversibly adsorbed to the air–water interface during stirring to form Pickering foam. According to the free energy adsorption formula, the larger the particle size, the greater the energy required for its instability [43]. Therefore, the foam stabilized by fraction 3 showed higher stability.

## 4. Conclusions

In this study, a sequence aqueous extraction method was designed, and five natural fractions rich in protein and polysaccharide were co-extracted from defatted flaxseed meal. The total protein recovery rate of this process exceeded 80%, which was more than the times higher than the reported method. The total extraction rate of defatted flaxseed meal was more than 90%. It was expected to realize the development of whole flaxseed meal. This method did not need the intervention of organic solvents as well as strong acid and alkali environment, conforming to the concept of green extraction. The composition, structure, and functional properties of the five fractions were also investigated. The protein content of the five fractions was between 29.71 and 73.05%. SDS-PAGE showed that the co-extraction would not cause damage to flaxseed protein, and the albumin could be enriched in fraction 4. The monosaccharide analysis exhibited that a large amount of flaxseed soluble polysaccharide was removed in the separation process of fraction 1, and the neutral polysaccharide content of fraction 1 was higher. The particle size distribution curve showed that fractions 1 and 5 have extra minor peaks representing aggregation. The compositional difference resulted in different functional properties. The order of solubility of the five fractions was as follows: fraction 4 > fraction 2 > fraction 1 > fraction 5 >> C-FPC > fraction 3, and among fraction 2 and 4 exceeded 90%. The EAI and ESI of fractions 1, 2, and 5 were higher than that of sodium caseinate, which could be used as high-efficient emulsifiers in the food industry. Polysaccharide could form a network structure in the continuous phase of the emulsion and promote the smoother interface of the droplets, which played a key role in the stability of the emulsion. The foaming property of fraction 5 enrich in globulin was the best; its FA and FS were 12.76 times and 9.89 times higher than C-FPC, respectively, which were similar to C-SPC. Thus, it is a promising foaming agent. The functional properties of fractions 3 and 4 were still far from commercial application, but exhibited certain potential. Thus, further modification will be the focus of follow-up research.

## Figures and Tables

**Figure 1 foods-12-01256-f001:**
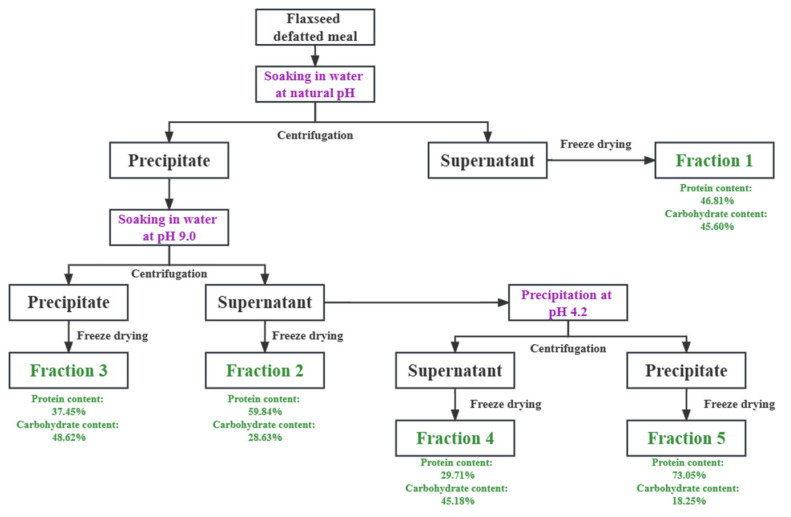
Schematic overview of the sequence aqueous extraction method for defatted flaxseed meal.

**Figure 2 foods-12-01256-f002:**
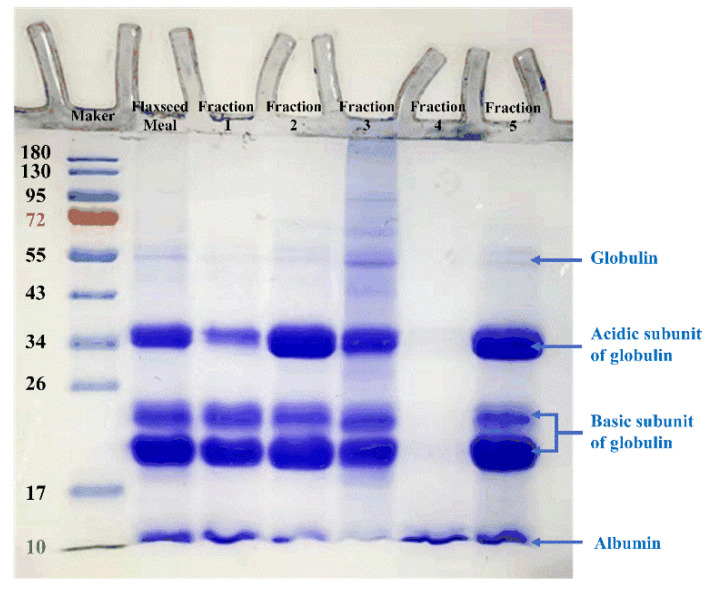
SDS-PAGE profiles of the defatted flaxseed meal and protein-polysaccharide natural fractions under reducing conditions.

**Figure 3 foods-12-01256-f003:**
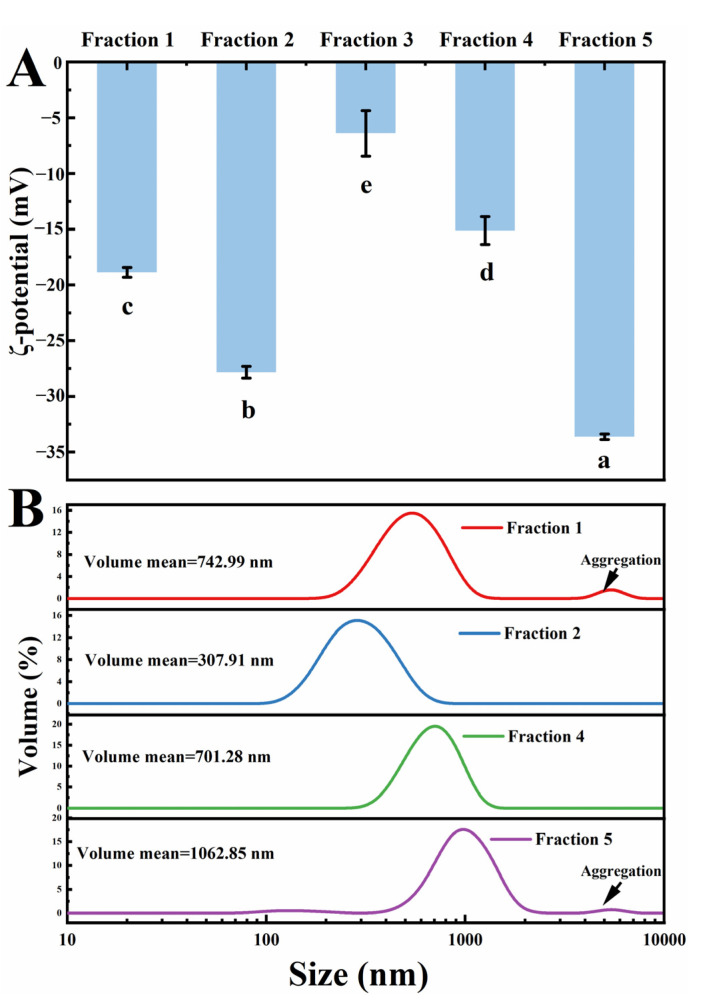
ζ-potential (**A**) and volume particle size distribution (**B**) of flaxseed protein-polysaccharide natural fractions. Comparisons were performed between values of the same serious and those without same letter(s) differed significantly at *p* < 0.05.

**Figure 4 foods-12-01256-f004:**
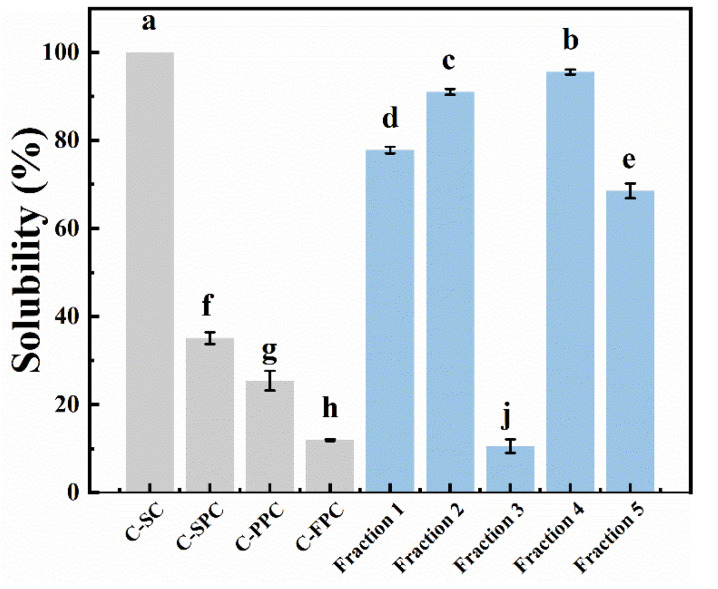
Solubility of flaxseed protein-polysaccharide natural fractions, C-SC, C-SPC, C-PPC, and C-FPC at pH 7.0. Comparisons were performed between values of the same serious and those without same letter(s) differed significantly at *p* < 0.05.

**Figure 5 foods-12-01256-f005:**
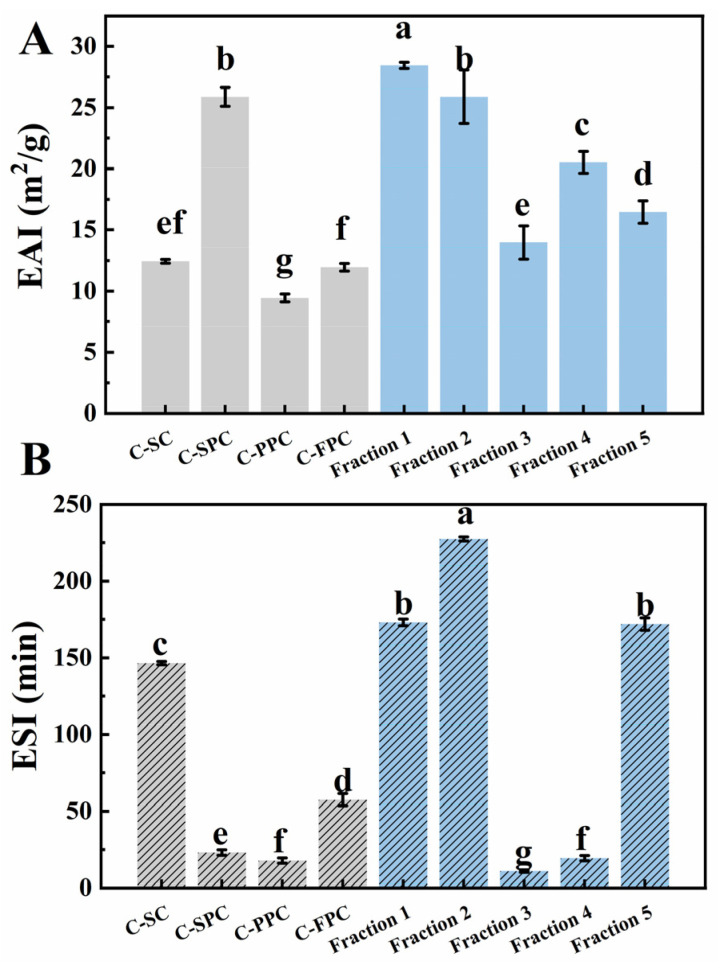
EAI (**A**) and ESI (**B**) of flaxseed protein-polysaccharide natural fractions, C-SC, C-SPC, C-PPC, and C-FPC. EAI and ESI represent the emulsifying activity index and emulsifying stability index, respectively. Comparisons were performed between values of the same serious and those without same letter(s) differed significantly at *p* < 0.05.

**Figure 6 foods-12-01256-f006:**
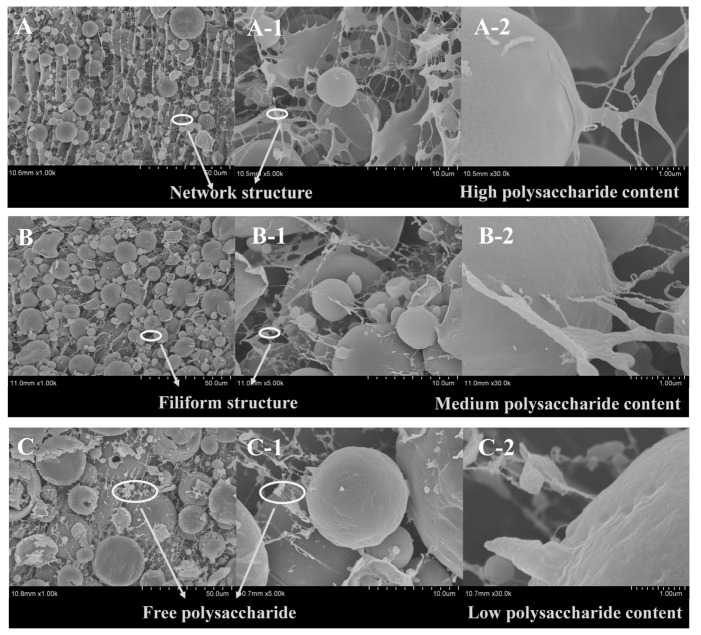
Cryo-SEM images of emulsions stabilized by fraction 1, fraction 2, and fraction 5 (**A**–**C**, ×1000; A-1, B-1, and C-1, ×5000; A-2, B-2, and C-2, ×30,000).

**Figure 7 foods-12-01256-f007:**
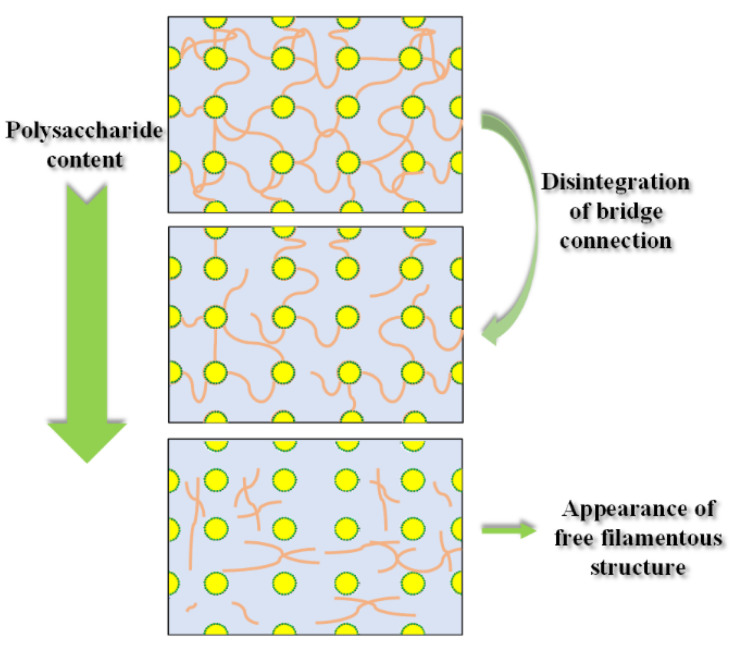
Schematic diagram of structural change of the continuous phase of emulsion with the decrease of polysaccharide content.

**Figure 8 foods-12-01256-f008:**
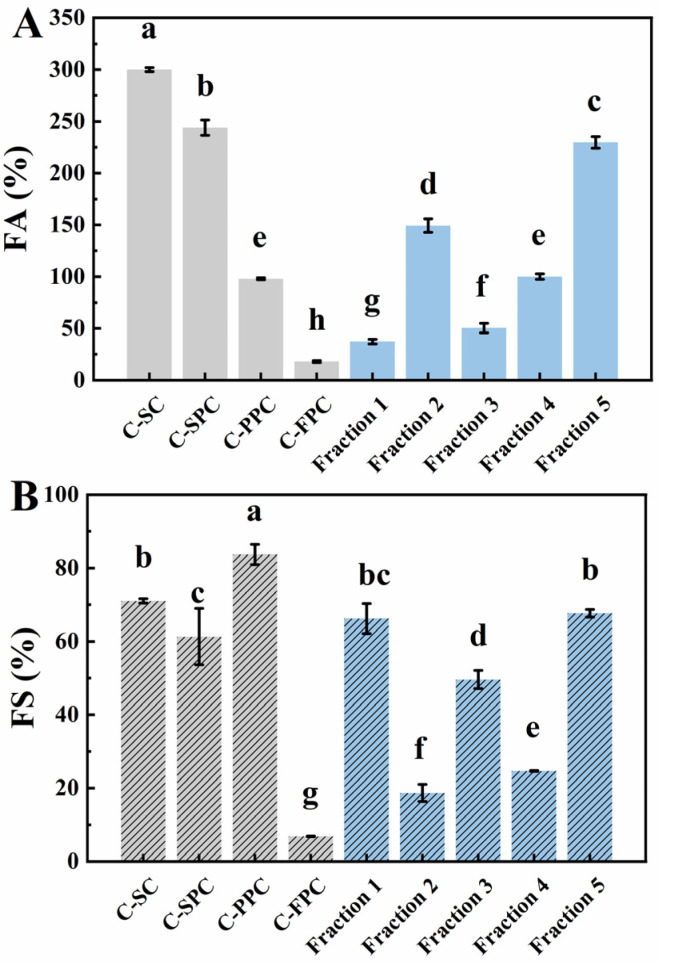
FA (**A**) and FS (**B**) of flaxseed protein-polysaccharide natural fractions and C-SC, C-SPC, C-PPC, and C-FPC. FA and FS represent the foaming ability and foaming stability, respectively. Comparisons were performed between values of the same serious and those without same letter(s) differed significantly at *p* < 0.05.

**Table 1 foods-12-01256-t001:** Composition, extraction yield, and protein recovery of defatted flaxseed meal and protein-polysaccharide natural fractions at a dry basis. Comparisons were performed between values of the same serious and those without same letter(s) differed significantly at *p* < 0.05.

Sample	Protein Content (%)	Carbohydrate Content (%)	Moisture Content (%)	Ash Content (%)	Protein and Carbohydrate Content (%)	Extraction Yield (%)	Protein Recovery (%)
Defatted flaxseed meal	41.74 ± 0.72	40.12 ± 0.35	11.15 ± 0.01	6.63 ± 0.06	82.26 ± 0.01	-	-
Fraction 1	46.81 c ± 0.59	45.60 b ± 0.30	2.02 d ± 0.19	5.57 c ± 0.12	92.41 a ± 0.89	28.44 b ± 0.52	29.82 b ± 0.89
Fraction 2	59.84 b ± 0.10	28.63 c ± 1.16	6.30 b ± 1.09	5.23 c ± 0.06	88.47 c ± 1.15	20.11 c ± 0.20	28.11 b ± 1.23
Fraction 3	37.45 d ± 0.40	48.62 a ± 0.56	4.16 c ± 0.35	9.77 b ± 0.15	86.07 d ± 0.26	45.23 a ± 1.21	38.50 a ± 1.19
Fraction 4	29.71 e ± 0.67	45.18 b ± 0.63	10.81 a ± 0.09	14.30 a ± 0.10	74.89 e ± 0.07	4.90 e ± 0.63	3.39 d ± 0.16
Fraction 5	73.05 a ± 0.15	18.25 d ± 0.32	6.46 b ± 0.13	2.23 d ± 0.12	91.31 b ± 0.20	10.28 d ± 1.16	17.19 c ± 1.48

“-” represents that data are unavailable or not measured.

**Table 2 foods-12-01256-t002:** Amino acids composition (mg/g protein) after hydrolysis of flaxseed protein-polysaccharide natural fractions.

Amino Acids (mg/g Protein)	Fraction 1	Fraction 2	Fraction 3	Fraction 4	Fraction 5
Histidine	22.96	20.02	34.89	16.81	28.45
Leucine	67.17	68.52	65.81	70.79	70.88
Isoleucine	65.25	97.41	56.01	91.22	63.58
Lysine	40.11	29.09	55.32	36.92	36.03
Methionine	45.43	81.19	28.63	82.24	47.94
Phenylalanine	51.33	41.18	61.13	27.29	59.11
Threonine	36.42	30.89	53.51	29.41	38.50
Valine	54.86	67.48	55.23	52.92	55.15
Alanine	38.31	34.02	52.19	25.96	45.50
Glycine	61.34	43.74	56.44	58.32	51.81
Proline	56.55	65.95	46.74	56.09	66.07
Serine	47.14	38.97	53.05	36.81	49.99
Glutamic acid	211.57	169.63	135.19	227.84	174.81
Aspartic acid	93.77	80.23	91.62	68.02	98.37
Arginine	65.11	91.98	101.05	85.22	74.14
Cystine	14.76	12.33	3.39	13.45	5.22
Tyrosine	27.90	27.37	45.40	20.72	34.45
Hydrophobic amino acids	378.91	455.76	365.74	406.50	408.24
Negatively charged amino acids	305.34	249.85	226.82	295.86	273.18
Aromatic amino acids	79.23	68.55	106.52	48.02	93.56
Positively charged amino acids	128.18	141.09	191.26	138.93	138.61
Sulfur-containing amino acids	60.19	93.53	32.02	95.70	53.16

**Table 3 foods-12-01256-t003:** Total carbohydrate composition (%, *w*/*w*) after hydrolysis of five flaxseed protein-polysaccharide natural fractions.

	Fucose	Rhamnose	Mannose	Arabinose	Galactose	Glucose	Xylose	Galacturonic Aicd	Glucuronic Acid
Fraction 1	2.97	16.37	0.00	10.56	17.01	28.25	19.74	5.09	0.00
Fraction 2	2.37	15.04	0.00	8.16	14.61	40.79	14.06	4.98	0.00
Fraction 3	1.34	0.00	1.34	27.00	14.35	25.46	24.16	6.06	0.28
Fraction 4	0.32	1.02	0.00	10.08	17.48	50.23	20.4	0.47	0.00
Fraction 5	3.78	26.32	0.00	5.95	12.28	34.85	8.34	8.48	0.00

## Data Availability

Not applicable.

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
