# Peer review of "Co-Extraction of Flaxseed Protein and Polysaccharide with a High Emulsifying and Foaming Property: Enrichment through the Sequence Extraction Approach"

_foods, 2023, doi:10.3390/foods12061256_

Round 1

Reviewer 1 Report

Dear Editor,

 Comments to Author:

1)    The biggest issue with this manuscript is that the extracting procedures are not well defined. As illustrated in Fig. 1, there no outgoing and omitting of other nutrients except protein from the solutions. Authors must explain that after each step of soaking in water, which part is protein and which parts consist of other nutrients such as carbohydrate or minerals of flaxseed, as there is obvious in  Schematic overview, both parts (supernatant and precipitate) have been used as flaxseed protein fraction and if no parts of other nutrients could be omitted by processing procedures applied in this trial, therefore, the soaking procedure will fail to concentrate the protein fractions of flaxseed.

2)    From the results given in Table 1, there is no trends for protein content from fractions 1 to 5. I the aim of the study is to concentrate and isolate the protein fractions of flaxseed, it is necessary to reduce the protein content in precipitate sub-fraction step by step and increase the soluble fraction of protein, but the data showing no trend in current trial.

3)    How the protein recovery calculated in the current trial. How was the similarity of amino acid content of recovered protein to the interval fractions? How was the destruction of the amino acids after each soaking step?

4)    In Table 2, why there is no data for amino acid content of defatted flax seed, which have been used for the original sample for processing? It is necessary to have data from the original sample and its similarity to amino acid content of recovered protein after processing.

5)    There is no significant differences between amino acid content of fraction 2 and fraction 4, why there is no considerable protein band in SDS-PAGE profiles of the flaxseed fraction 4 and other parts?

6)    Table 3, it is necessary to have data for total carbohydrate composition of defatted flaxseed meal to compare the total carbohydrate composition in fractions and original sample.  

7)    In general, the structure of the introduction and discussion are not well organized and is not more specialized about the effects of processing condition on outcomes of recovered products and what is the advantages and disadvantages of these method. Discussion section in this trial is too short and it is not appropriate to measured parameters.

Reviewer 2 Report

The manuscript has investigated the emulsifying and foaming properties of flaxseed protein and polysaccharide by co-extraction method. The topic is interesting; However, the manuscript has several problems, mainly:

1. Section 2.1; Protein contents of 83.23, 80.47 and 83.39% are generally considered protein concentrates, not protein isolates.

2. Section 2.7; Check the title. Also, explain more details about the experiment.

3. L 222. ; What did the authors mean by "functional properties"?

4. Check the stale of the citations. For example, L 274, L 436, etc.

5. Section 3.4; Check the title.

6. Section 3.6.1, Why are the solubility results only compared to C-FPI?

Reviewer 3 Report

This is a good piece of work. There are some few suggestions with the purpose of improving this report.

Abstract. It is not clear here how Fraction 5 was obtained. Later on, in R and D one learns with the diagram about the whole matter. But improve Abstract.

Table 1. Indicate if contents are in dry / wet basis.

Table 2. Very strong variations in amino acids content in five fractions. Interesting aspect; please try to describe clearly this important behavior.

Figure 2. I see four main fractions not five. Instead of SDS-PAGE authors should use, additionally, a better technique for protein characterization. Please consider this aspect.

Fraction 3 solubility. It is extremely low. It would be interesting to know the reasons of such trend; the scientific reasons. 

Conclusions. They are rather poor especially because of the importance to state what type of research is needed based on the results obtained in this work. Please try to improve this section.

Figure S1. Where is Figure S1 on protein denaturation by DSC ? 

Round 2

Reviewer 1 Report

revision and response to the previous comments are acceptable and no other comments is needed.